# Tensile Strength of Class F Fly Ash and Fly Ash with Bentonite Addition as a Material for Earth Structures

**DOI:** 10.3390/ma15082887

**Published:** 2022-04-14

**Authors:** Mariola Wasil, Katarzyna Zabielska-Adamska

**Affiliations:** Faculty of Civil Engineering and Environmental Sciences, Bialystok University of Technology, Wiejska 45E Street, 15-351 Bialystok, Poland

**Keywords:** fly ash, clay, tensile strength, barrier deformation, Brazilian test, direct tensile test

## Abstract

The behavior of soils under tensile stress is of interest to geotechnical engineers. Tensile strength of soils is often associated with tensile fractures that can generate a privileged flow path. The addition of bentonite improves the plastic properties of the soil, therefore the study was conducted for the compacted class F fly ash and fly ash with various bentonite additions. An amount of bentonite was: 5, 10 and 15%, calculated in weight relation to dry mass of samples. The tensile strength of compacted clay was also established, for comparison. Laboratory tests were carried out using the direct method (breaking) on cylindrical samples and the indirect method (the Brazilian test) on disc-shaped specimens. For this purpose, a universal testing machine with a frame load range of ±1 kN was used. It is stated that bentonite considerably influences the tensile strength of the fly ash evaluated with both methods. The tensile strength values obtained with the Brazilian method are comparable or higher than those obtained with the direct method. The achieved tensile strength values of compacted fly ash, improved by 10−15% of bentonite addition, are comparable with the results obtained for clay used in mineral sealing, while the strain at maximum tensile strength is similar in the direct test and lower in the indirect test.

## 1. Introduction

The behavior of soils under tensile stress is a topic of great importance interest for geotechnical engineers [1]. The tensile strength of soils is much often associated with tensile cracks, which can occur in various earthworks, such as embankments, dams, slopes, reinforced soil retaining walls (MSE), when the structure loses its stability. The tensile strength is also significant in the case of mineral sealing layers under embankments and landfills or their covers. Uneven settlement can initiate destruction or cracks of the sealing layer and generate a privileged flow path.

The failure is initiated by formation and propagation of cracks triggered by tensile stress, particularly when this stress reaches or exceeds the tensile strength in the soil layer. The hardiness of the soil to uneven settlement can be characterized by the extension or breaking strength results and the deformation of the soil specimen. It is assumed in geotechnical practice that the tensile strength of the soil is insignificant or equal to zero, as it is a relatively small value in comparison with the soil compressive strength. It should be noted, however, that in order for fly ash specimen to crack, it is necessary to overcome the resistance to structural cohesion caused by the interlocking of the crushed spherical ash grains [2] and, *inter alia*, capillary stresses (e.g., [3]) in compacted fly ash.

Methods for evaluating the soil tensile strength have not been developed (except extension test) in soil mechanics, and it is required to adjust the methods of related scientific disciplines and specialties. Laboratory tests of tensile strength can be performed using direct and indirect methods. The direct methods are applied to carry out an extension test in triaxial compression conditions, as well as a breaking cylindrical sample inserted into two-part molds [4,5] or specimens in the shape of a dog-bone [6,7]. The test can be also carried out on hollow cylinder subjected to various internal and external hydrostatic pressures [1] or on annular sample tension-loaded on the inner hole [8,9]. The direct tests are as well conducted with specially designed direct tension apparatus, equipped with a bipartite box of various shapes [10,11,12], and by model test of sealing layers in the centrifuge [13]. In the indirect method, the so-called the Brazilian method, cylindrical or disc specimens are compressed alongside length [7,14,15,16]. During testing with the Brazilian method, the following solution of applying force to the sample can be used [17]—by flat loading platens, platens with small diameter steel rod, platens with cushion, and curved loading jaws. In other indirect method, a rectangular sample or supported in five points beam is bent [18,19]. The results of the research presented in the literature concern only one research method, so it is difficult to draw a conclusion about the comparability of the presented research results.

Tensile tests have been most often performed to study the tension properties of compacted clay built-in core of the earth dams or a system of landfill sealing layers. The literature emphasizes that the most important thing is achievement maximum extension without cracking. The drying clay samples develop an important shrinkage that affects the water retention and tensile strength properties [20,21]. However, very limited information on fly ash tension has been published so far. Mollamahmutoğlu and Yilmaz [14] tested fly ash by the Brazilian method and obtained tensile strength equaled 7.8–10.9 kPa. Ghosh and Subbaro [22] tested class F fly ash stabilized with lime alone or in combination with gypsum. Specimens of fly ash and fly ash with 4–6% lime did not show the Brazilian tensile strength. The Brazilian strength of the stabilized fly ash improved with an increase in lime content and curing period.

The aim of the paper is to determine the tensile value assessed by the tensile strength and deformation value that the material can transfer without cracking. The tensile strength was established for compacted fly ash and fly ash samples with the bentonite addition improving the fly ash plasticity, without affecting, or even with improving mechanical properties of fly ash [23,24,25]. The presented results of laboratory tests were performed using the direct and indirect method. It was found that the Brazilian test may overestimate the tensile strength results of the granular materials tested. The test results will be compared with the results obtained for mineral cohesive soil, used to construct the sealing layers, as a soil characterized by tension properties.

## 2. Materials and Methods

### 2.1. Materials

Studies were performed on the fly ash and bottom ash mixture, henceforth referred to as fly ash because there was only a vestige of bottom ash in the mix. This fly ash was a by-product of bituminous coal combustion at the Thermal-Electric Power Station in Bialytsok, Poland. It was taken from the dry storage yard. Considering the coal type and main components of the fly ash (Table 1), according to ASTM C618-19 [26], it should be qualified as *class F*. It corresponds to the grain size of sandy silt (saSi) [27]. The fly ash was tested alone and as a mix with 5, 10 or 15% of commercially available bentonite (clay − Cl), calculated in weight relation to dry mass of samples. The main chemical elements building the structure of tested bentonite were: silica (Si)—31.13%, aluminum (Al)—7.67%, calcium (Ca)—1.92%, sodium (Na)—1.54% and magnesium (Mg)—0.97%.

Comparative tensile test on compacted high plasticity soil—sandy silty clay (sasiCl according to EN ISO 14688-1 2017 [27]), used for the erection of mineral barriers, has been performed too. Figure 1 shows the grain-size distribution curves of all tested materials.

The physical parameters for the averaged research samples: median grain diameter (*D*_50_), density of solid particles (*ρ*_s_) and specific surface (*S*_s_) measured by the methylene blue spot test, are shown in Table 2.

All tested samples were compacted by the standard Proctor method (SP) at optimum water content (*w*_opt_) to the dry density corresponding to the value of maximum dry density (*ρ*_d max_), allowing only a single compacting of the same fly ash sample [2]. Bentonite was added to fly ash immediately prior to compaction, and then the samples were mixed and compacted. Samples were assessed at room temperature directly after compaction and after 7 and 28 days of remaining in a humidity chamber at moisture content above 96%. The tensile tests were repeated 2–4 times for each case. Compaction curves of all tested materials are shown in Figure 2.

### 2.2. Methods

Laboratory research of tensile strength has been achieved by means of electromechanical universal testing machine with a frame load range during extension/compression ± 1 kN. The research was performed using both the direct and indirect methods (Figure 3).

The samples were compressed or extended applying a force displacement monitored with the optical transducer during which the compressive or extending force dependency on the deformation of the test zone was recorded. The speed of displacement increment Δ*l* was 1.2 mm/min = 0.02 mm/s. During the experiment compressive or extending force and the momentary length of the specimen were assessed. These values enable the determination of the state of stress and strain in the specimen.

#### 2.2.1. Direct Method

Tensile test was performed on cylindrical samples of 36 mm in diameter and 82 mm in height. Specimens were prepared by compaction in a bipartite device constructed for non-cohesive soil sample forming for tests in triaxial apparatus. Then the samples were carefully placed in especially designed two-part molds, allowing the sample to be deposited in the testing machine (Figure 3b) and subjected to extension (breaking).

The stress causing a noticeable material plastic deformation is called its yield stress. Tensile strength is defined by yield stress, also known as engineering stress, calculated based the relationship (1):(1)$$\sigma =\frac{F\left(t\right)}{S_{0}}$$
or a true stress (2):(2)$$\sigma =\frac{F\left(t\right)}{S\left(t\right)}$$
where: *F(t)* is the value of tensile force in time *t*, *S*_0_ is the cross-sectional area of sample prior to test, and *S(t)* is the instantaneous cross-sectional area of specimen in time *t*.

Analogously, engineering strain is stated by a Formula (3):(3)$$\varepsilon =\frac{l\left(t\right)-l_{0}}{l_{0}}=\frac{\Delta l\left(t\right)}{l_{0}}$$
and a true strain (4):(4)$$\varepsilon =ln\frac{l\left(t\right)}{l_{0}}=ln\frac{l_{0}+\Delta l\left(t\right)}{l_{0}}$$
where: *l(t)* is the instantaneous length of specimen in time *t*, *l*_0_ is the length of sample prior to test.

The tensile strength is specified as the maximum tensile stress the material can carry. In the case of yielding material the stress reaches a maximum value prior to its fracture and relates to the point at which necking initiates. After achieving the tensile strength, the material shows a decrease in the tensile force until the specimen fractures at a point relating to a breaking force. Breaking stress is associated to the smallest cross-section of an extended specimen. Less plastic material cracks before necking begins. Very brittle material fractures ahead of its deformation. This type material has tensile strength, but no yield stress.

#### 2.2.2. Indirect Method

Tensile test was achieved using the Brazilian splitting tensile strength test conducted by compressing a cylinder at side length. The dimensions of the sample were determined in accordance with ASTM D3967-08 [28], where the preferred relation of the height of the cylinder (disc) to its diameter was given as H/D = 0.2–0.75. The diameter of the specimen should be at least 10 times greater than the biggest grain in the tested material. These relations were found in the disc sample of 65 mm ID and 20 mm high, compacted in a bipartite device constructed for non-cohesive soil specimen forming for oedometric tests. Loading of the specimen perpendicularly to its axis triggers cracks along the plane passing through the cylinder axis, generated primarily by tensile forces, leading to the sample splitting (Figure 3c). Loading was provided by flat loading platens.

The tensile stress in the Brazilian method is defined by the formula:(5)$$\sigma =\frac{2F_{max}}{\pi HD}$$
where: *F*_max_ is the peak value of compression force at failure, *H* is the height of disc sample (the thickness), *D* is the diameter of disc sample.

Deformation in the Brazilian method is described as a vertical displacement (reduction of the diameter of the sample) or relative strain:(6)$$\varepsilon =\frac{H-H_{0}}{H_{0}}$$
where: *H* is the length of sample at fracture, *H*_0_ is the length of sample prior to test.

As stated above, loading the sample leads to the splitting of the sample. In the older literature, there is a prevailing view that in accordance with the criterion of Griffith’s, the destruction begins in the middle of the sample. On the other hand, more up to date literature positions state that it is the stress concentration at points of load contact that can lead to the destruction of the samples [17]. Authors stated based on their own research that during the Brazilian test, both forms of destruction propagation may occur—from the center of the specimens and the contact load points [5].

## 3. Results and Discussion

Figure 4 presents a selected results of tensile strength tests of the fly ash and fly ash with 5%, 10% and 15% addition of bentonite compacted at optimum moisture content. The charts show tensile results as the dependence of stress: extension (breaking)—direct method (Figure 4a–c) or compressive—the Brazilian method (Figure 4d–f) on the relative strain during the test. The specimens are characterized by relative strain increasing proportionally to the bentonite addition, which, however, in the case of the direct method, do not exceed 0.6%. During the indirect (Brazilian) tests, the strain also increased with the plasticity of the specimens, not exceeding 1.4%, referring to all tested samples.

Samples tested by the indirect method do not reveal the elongation from which strain is calculated. When the maximum compressive force was reached, the vertical displacement caused the load-transferring surfaces to move closer together, causing the sample to compress. In the case of the direct method, once the maximum tensile force is exceeded, the vertical displacement corresponds to an increase in the distance between the two parts of the torn sample.

From Figure 4, it can be concluded that the curing time of the sample clearly affects the increase in tensile strength, regardless of the test method (direct or indirect). The tensile strength of samples tested immediately after compaction by both methods has similar values, although in the direct method the maximum value of the tensile force is reached at a lower relative strain value.

Table 3 summarizes the values of maximum stresses obtained in the direct and indirect tests of the fly ash with bentonite addition. The peak tensile stress is the tensile strength of the sample at failure, which is connected with the maximum tensile stress required to counterbalance the contact force between the particles along the failure plane [29].

It can be observed based on Table 3 that the bentonite addition affects the tensile strength of fly ash tested with both methods. It relates to rising the plasticity and dry density of compacted samples (see Figure 2), which agrees with Ibarra et al. [30] test results of silty clay. In general, the higher values are obtained with the indirect (Brazilian) method—from 7.09 kPa to 80.82 kPa—than with the direct method, where the mean value of σ varied from 4.96 kPa to 37.17 kPa. The results of the Brazilian test may overestimate tensile strength values due to biaxial stress instead of tension condition. It indicates that the Brazilian tensile testing cannot replace direct tensile testing, as Li and Wong [17] also stated. Ibarra et al. [30] claim that the indirect compression test is not applicable to the cracking of soil in a continuous body. Referring to Table 2, the relative strain value for the direct method range from 0.1% to 0.6% and the indirect method from 0.3% to 1.4%.

Mollamahmutoğlu and Yilmaz [14] obtained the tensile strength of fly ash in the Brazilian test equal to 7.8–10.9 kPa. For fly ash, improved by 10−30% bentonite, they also received considerably lower strength values—25.7–88.3 kPa compared to the results presented in the study. The deformation level was determined by them at 1% for all presented tests.

Curing time also affects the tensile strength of tested samples. The values obtained directly after compaction, in the case of the direct method, are from 8.35 kPa to 13.43 kPa, after 7 days of curing from 14.05 kPa to 24.02 kPa, and after 28 days of curing from 4.96 kPa to 37.17 kPa. In the case of the Brazilian test, the tensile strength of samples tested immediately after compaction is from 7.09 kPa to 14.66 kPa. Samples cured 7 days obtained values from 11.32 kPa to 38.72 kPa and cured 28 days—from 7.68 kPa to 80.82 kPa. Time of curing samples had no significant effect on the strain value at the time of failure. The widest range of strain is obtained for the indirect method for samples tested after 28 days of curing, from 0.3% to 1.4%.

The largest differences in tensile strength test results are found in samples cured for 28 days, greater for the indirect method than for the direct method, which may be due to the curing of specimens and their dimensions—smaller in the indirect method. Figure 5 shows the statistical values of the tensile strength tested using the direct method—by breaking and the Brazilian indirect method.

Then, the matrices of linear correlation coefficients *r* were calculated. Missing data were replaced with mean values for the specified percentage of bentonite addition. Statistical calculations gave satisfactory results in the case of the test results obtained with the Brazilian method. For 14 analyzed cases of strength determined by the Brazilian method, in all cases the correlation coefficients *r* are statistically significant at the significance level of α = 0.05 (*r* = 0.87–0.99). The highest values of the linear correlation coefficients *r* are obtained for the samples cured for 7 days (for the dependent variable σ the value of the correlation coefficients *r* = 0.99). Regarding the test results obtained by the direct method, satisfactory results were not obtained—in 13 analyzed cases, statistically significant correlation coefficients *r* are obtained at the significance level α = 0.05 only in the case of samples cured for 7 days, where the values of the linear correlation coefficients *R* for the dependent variable σ is 0.60.

In multiple linear regression, the content of bentonite addition is used as the independent variable. The analysis of the residuals of all cases show, that in the case of the values obtained both by the direct and the direct method, there are no outliers. All the rest are within ± twice the standard deviation. Table 4 presents the results obtained during the multiple linear regression analysis for particular curing times. Statistically significant results are obtained for all samples cured with the Brazilian method, and for the direct method (breaking)—only for samples cured for 7 days.

To provide the comparison of the tensile strength of fly ash with various bentonite additions to mineral cohesive soil, Figure 6 has been presented. The results are shown for FA + 10%B and FA + 15%B after different curing times and for the compacted sandy silty clay (sasiCl), tested by both methods. Charts are presented as the dependence of tensile stress on the relative strain.

In the case of the direct method (Figure 6a,b), the highest tensile strength value is achieved for the FA + 10%B sample cured for 28 days. The FA + 15%B samples tested by the direct method, compare to the compacted sasiCl samples, demonstrate lower values of tensile strength.

In the case of the indirect method (Figure 6c,d), the compacted sandy silty clay reach its maximum tensile strength value at a much higher relative strain (about four times) than the fly ash with various bentonite additions. For the direct method, the value of relative strain in time the maximum tensile stress is reached is similar for all tested materials. The value of tensile strength of compacted sasiCl tested by the Brazilian method significantly exceeds the values obtained for samples of FA + 10–15%B tested immediately after compaction and after 7 days of curing, while samples tested after 28 days of curing have values similar to the results obtained for sandy silty clay.

In the case of mineral cohesive soil, for selected samples presented in Figure 6, the tensile strength reaches value 98.04 kPa tested by direct method and 39.60 kPa tested by indirect method. During the direct test on silty sandy clay samples, the necking of the sample was observed, before achieving the peak value of tensile stress, similar to the Vaniček research [1]. The clay tested by Wang et al. [31], which had a lower dry density value (from 1.60 g/cm^3^ to 1.75 g/cm^3^), brittle failure was observed. For fly ash, even the addition of bentonite in the amount of 15%, does not result in the necking of the sample during the test—the failure of the sample is brittle. Thus, during the tensile test of the fly ash using the direct method, the real stress is equal to the engineering stress due to the constant cross-sectional area of the specimen during the test. In the majority of the tests, a convergence of the tensile strength limit and the point corresponding to the destruction of the specimen is observed, proving its high brittleness.

## 4. Conclusions

Based on conducted tests of tensile strength, the following conclusions can be drawn:(1)Bentonite considerably influences the tensile strength of fly ash, established by both direct and indirect (Brazilian) methods. The addition of 10% bentonite enhances the tensile strength of fly ash compacted by the standard Proctor method at optimum moisture content, tested by the Brazilian method, by more than 10 times.(2)Raising the bentonite content to 15% increases the strain recorded at sample fracture, while the strength increase remains the same or decreases. Addition of bentonite increases the relative strain of the sample in proportion to the percentage of the additive. The value of relative strain during cracking in the fly ash sample with 15% addition of bentonite was established as: 1.4% with the Brazilian method and 0.6% with the direct method.(3)The tensile strength values obtained by means of the Brazilian method are comparable or higher than the direct method, which is especially apparent at higher values of destructive forces. Results of the Brazilian test may overestimate the tensile strength of tested granular materials.(4)The curing time of the fly ash with various bentonite additions affects the tensile strength tested by direct and indirect methods. The lowest values of the tensile strength are obtained for the samples tested directly after compaction, the highest—after 28 days of cure.(5)The tested cohesive soil, sandy silty clay (sasiCl) shows a maximum value of tensile stress at higher relative strain—about four times—than fly ash with bentonite addition tested by the direct method. In the case of the indirect test method, the relative strain while achieving maximum tensile strength is comparable between cohesive soil and fly ash with bentonite additions. The values of maximum tensile strength are comparable for tested clay and the fly ash improved by 10–15% of bentonite addition after 28 days of curing.(6)According to the results obtained from the tensile tests, it can be assumed that the addition of 10% bentonite in the case of fly ash allows to obtain optimal values of tensile strength. Increasing the bentonite addition to the fly ash to 15% slightly increases the tensile strength.(7)The tests carried out and presented in the paper have shown that different tensile strength values can be obtained depending on the method. This suggests that the test conditions (methods) should be always indicated or described with the presented value of the tensile strength of the soil.

## Figures and Tables

**Figure 1 materials-15-02887-f001:**
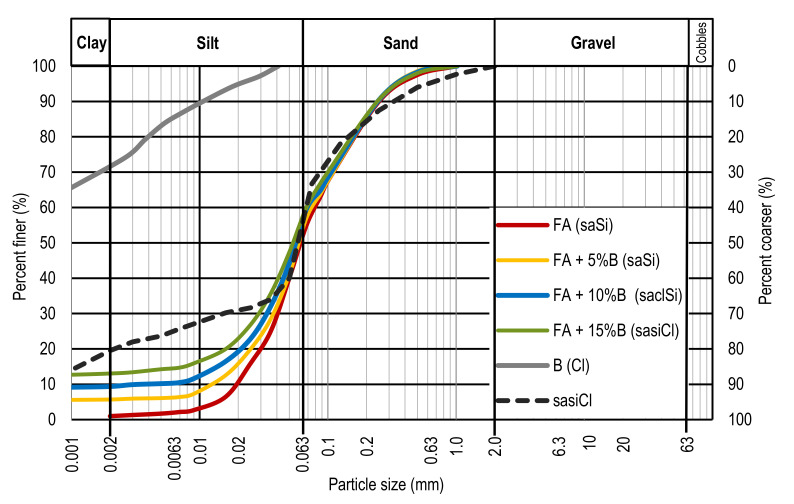
Grain-size distribution curves of all tested materials.

**Figure 2 materials-15-02887-f002:**
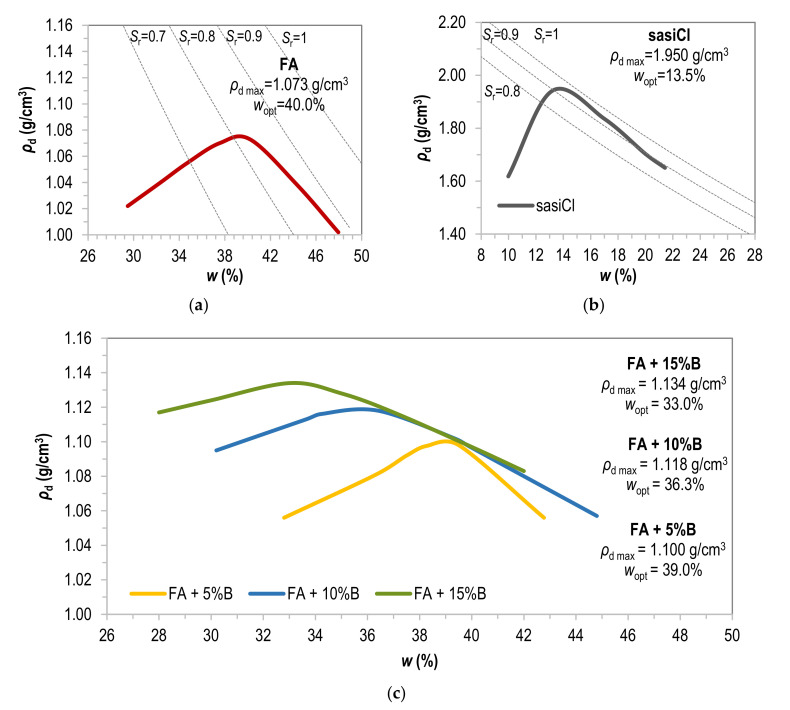
Compaction curves of: (**a**) fly ash without additives; (**b**) sandy silty clay; (**c**) fly ash with the addition of 5, 10, and 15% of bentonite.

**Figure 3 materials-15-02887-f003:**
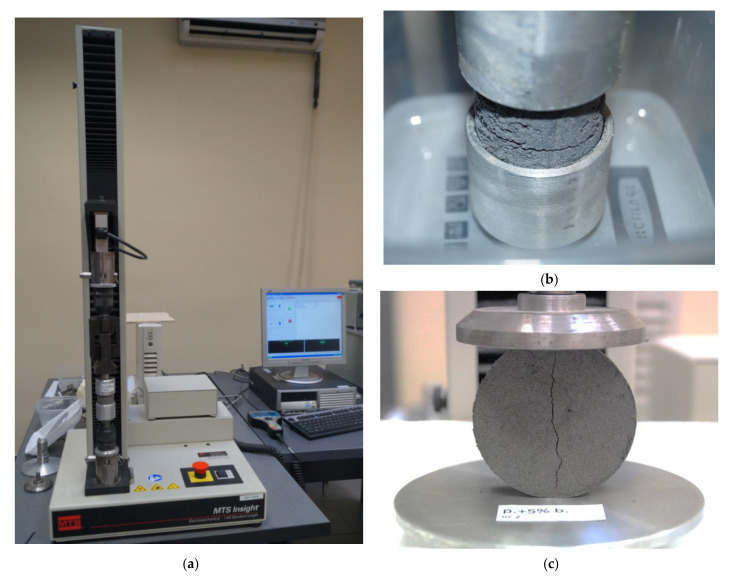
Laboratory testing of the tensile strength of the soils: (**a**) MTS Insight testing system; (**b**) the view of the sample during direct test; (**c**) the view of the sample during indirect test.

**Figure 4 materials-15-02887-f004:**
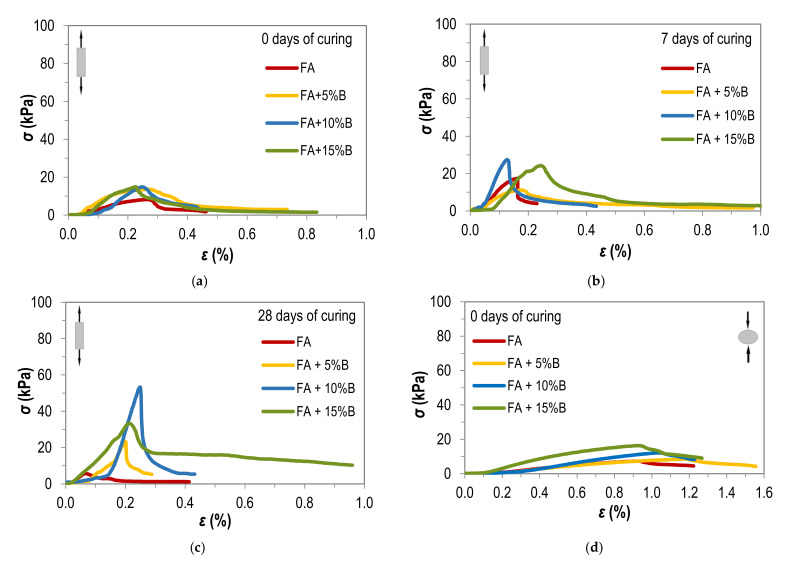
The results of laboratory tests of the tensile strength of the soils after different time of sample curing for: (**a**–**c**) direct test; (**d**–**f**) indirect test.

**Figure 5 materials-15-02887-f005:**
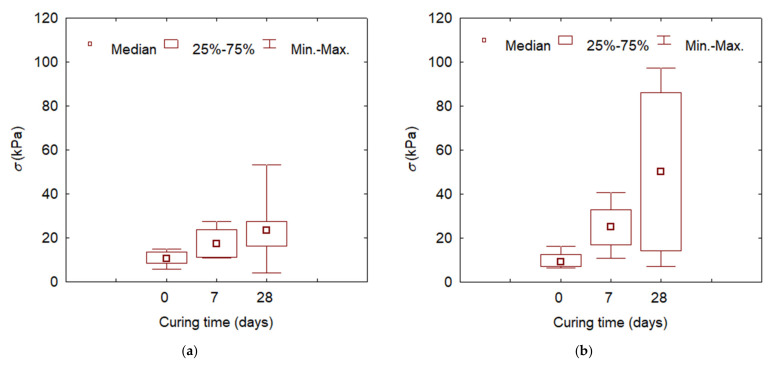
Ranges of variation of tensile strength values with time of curing: (**a**) direct method; (**b**) Brazilian method.

**Figure 6 materials-15-02887-f006:**
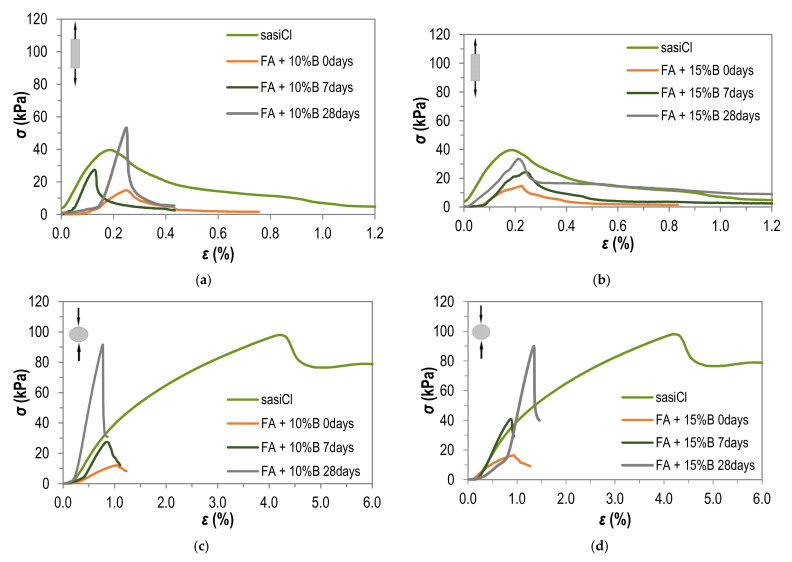
Tensile strength of sandy silty clay (sasiCl) and fly ash with 10% (FA + 10%B) and 15% (FA + 15%B) addition of bentonite tested after different curing times by: (**a**,**b**) direct method; (**c**,**d**) indirect method.

**Table 1 materials-15-02887-t001:** The main components of tested fly ash.

Designation	Content (%)
Si as SiO_2_	46.42
Al as Al_2_O_3_	21.01
Fe as Fe_2_O_3_	6.27
Ca as CaO	3.12
Mg as MgO	0.87
S as SO_3_	0.68
P as P_2_O_5_	0.092
Ti as TiO_2_	1.4
Mn as Mn_3_O_4_	0.070
Na as Na_2_O	0.48978
K as K_2_O	0.63535
C as a loss of ignition	8.23

**Table 2 materials-15-02887-t002:** The parameters of tested materials.

Material	D_50_ (mm)	*ρ*_s_ (g/cm^3^)	*S*_s_ (m^2^/g)
FA	0.061	2.18	21.01
FA + 5%B	0.059	2.18	42.24
FA + 10%B	0.056	2.22	57.90
FA + 15%B	0.054	2.24	67.73
sasiCl	0.058	2.68	102.58
B	–	2.45	506.84

FA—fly ash, B—bentonite, sasiCl—sandy silty clay.

**Table 3 materials-15-02887-t003:** Mean values of tensile strength of the tested samples.

Material	Days of Curing	Method	Number of Samples (pcs.)	Range of *σ* (kPa)	Mean *σ* (kPa)	Range of *ε* (%)	Method	Number of Samples (pcs.)	Range of *σ* (kPa)	Mean *σ* (kPa)	Range of *ε* (%)
FA	0	Direct	2	8.15–8.55	8.35	0.3–0.5	Indirect	2	6.86–7.32	7.09	0.9
FA + 5%B			2	13.26–13.56	13.41	0.3–0.4		2	6.39–8.45	7.42	1.0–1.2
FA + 10%B			4	5.80–14.83	10.71	0.2–0.3		2	10.10–12.11	11.10	0.9–1.0
FA + 15%B			3	9.73–15.92	13.43	0.2–0.4		2	12.99–16.34	14.66	1.0
FA	7		2	10.71–17.39	14.05	0.1–0.2		2	10.98–11.65	11.32	0.7–1.0
FA + 5%B			4	11.20–24.95	16.08	0.1–0.6		2	21.96–22.78	22.37	0.6–0.7
FA + 10%B			3	11.30–27.41	20.27	0.1–0.3		2	27.42–28.07	28.20	0.7–0.8
FA + 15%B			2	23.78–24.27	24.02	0.2–0.3		2	36.60–40.83	38.72	0.7–0.8
FA	28		2	4.13–5.80	4.96	0.1		2	7.22–8.14	7.68	0.3–0.4
FA + 5%B			3	12.97–24.36	20.24	0.1–0.3		4	20.67–47.63	35.91	0.7–1.2
FA + 10%B			3	27.50–53.25	37.17	0.2–0.5		4	56.39–97.32	79.43	0.8–0.9
FA + 15%B			3	16.31–33.50	22.30	0.2–0.4		4	64.95–90.00	80.82	0.9–1.4

FA—fly ash, B—bentonite.

**Table 4 materials-15-02887-t004:** Results of the multiple linear regression of the dependence of the tensile strength on the bentonite addition (0, 5, 10, 15%) for different curing times.

Test Method	Curing Time (Days)	*R* ^2^	Adjusted *R*^2^	Standard Error of Estimation	*F*-Test	Statistical Relevance
**Direct**	0	0.1043	0.0228	2.9994	1.2803	irrelevant
	7	0.3625	0.3045	5.0850	6.2545	relevant
	28	0.2514	0.1834	12.0791	3.6942	irrelevant
**Indirect**	0	0.8628	0.8514	1.2652	75.4958	relevant
	7	0.9742	0.9721	1.5667	453.7332	relevant
	28	0.7524	0.7318	16.2163	36.4687	relevant

## Data Availability

The data presented in this study are available on request from the corresponding author.

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
