# Peer review of "Tensile Strength of Class F Fly Ash and Fly Ash with Bentonite Addition as a Material for Earth Structures"

_materials, 2022, doi:10.3390/ma15082887_

Round 1

Reviewer 1 Report

The manuscript (ms) focused on tensile strength evaluation on structure made of fly-ash. This study, according to the authors, is underdeveloped. Therefore, for enhancing the quality of this ms, this study should answer some questions below:

  1. What is the importance to compare between direct and indirect methods? This was missing in the introduction.
  2. Since the authors emphasized on fly-ash, particle size and the distribution of this material should be written in text, not just showed the curve (p.2, l.86)
  3. Make consistent remarks of Figure 1. sasiCl or saClsi? It is better to write sa-si-cl since it is an abbreviation of sandy-silt-clay. Cl would be chloride in term of chemical.
  4. Since this study was developing method, UTM (Universal Testing Machine) used should be explained its identity (merk, made in, capacity, loading used).
  5. Further, authors mentioned using frame loading (in method section). Could the authors show the figure/ picture of this (at least in supplementary)? UTM usually applied one point loading or three points loadings for mechanical testing.
  6. Why the authors chose cylindrical structure for this testing? Generally, for tensile strength, the dog-bone shape is usually used. Reason of this was missing in the text.
  7. The direct method samples seemed like a tension sample. Could the authors explain more? Since this is “a new method” preparation of the sample must be clear. Illustration could be helped.
  8. Why the authors used different replication in number of samples? In this case 2-4 replications. Usually, at least 3 replications were applied when using statistical analysis. Duplo can be applied if the source of materials was very rare such as in chemistry field. In this case fly ash was abundant.
  9. In the results and discussions, statistical analysis results were appeared. Why in the method the statistical analysis was missing? Authors should add them.
  10. Since the bentonite was only 5-15%, in my opinion it was better to say “addition” rather than “mixture” (in abstract and material sections).
  11. It is better to consistently mention “curing” rather than “day of care” (Table 2, Figure 5).
  12. In the conclusion, the authors should suggest the best addition of bentonite for structural purpose, for instance at 10%.

Author Response

The authors would like to thank the Reviewer for constructive and insightful review and comments in relation to this work. Please see the attachment for the responses.

Reviewer 2 Report

Authors present results from experiments on the tensile strength of fly ash reinforced by sodium bentonite using direct and indirect methods of laboratory measurements. The work is interesting, well analyzed and accessible to a general readership. The topic falls definitely within the scope of the special issue, especially the characterization aspect. Subject to addressing the questions and comments raised below the manuscript would be publishable in Materials.

) In the results it would be interesting to include as an intermediate time of curing 2 weeks (14 days). This would further validate the trends observed between 1 and 28 days.

) Title seems too generic. It would be good to include the reinforcement type.

) Legends in table should be extended to explain the presented quantities (for example Table I).

) Authors could present the results of the direct and indirect tests under the same scales to aid visual comparison as they report the same quantities. For example, the direct test results could have their y-axes from 0 to 100 in Fig. 4. Same in Figure 6.

) The manuscript has some syntax problems, mainly by referring to past tense when presenting results of the current work. For example:

Abstract (line 12): “it was stated” -> “it is stated” (but it can be removed as well without loss); “influenced” -> “influences” etc.

) Some minor grammar/syntax errors:

Authors use in many instances “-“ to separate the sentences. Using of semi column should be preferred.

Line 22: “of great importance interest of” -> “of great importance for”.

Line 196: “which corresponds” -> “which agrees”.

Author Response

The authors would like to thank the Reviewer for constructive and insightful review and comments in relation to this work. Please see the attachment with the answers.

Reviewer 3 Report

In this paper, the tensile strength of compacted fly ash and fly ash bentonite mixtures was determined by indirect and direct methods. The paper is well-written and interesting. Many thanks to the authors for their hard work, but it is not enough to do some comparative experiments, it is necessary to draw some fundamental conclusions, not simply to list the experimental data.The authors need to improve their manuscript based on the comments below. 

  1. Why this paper chose fly ash as the main body to add bentonite study instead of bentonite as the subject to add fly ash?
  2. Abstract needs improvement. Start with some general introduction, follow with objectives and some major conclusions.
  3. You need some general conclusions, rather than repeating your results.
  4. What are your objectives?
  5. line212. Why was the tensile strength for a curing time of 14 days not recorded?
  6. Why do we choose the direct method and the indirect method to study the tensile strength separately in this paper? What conditions are more suitable for the direct or indirect method of measuring tensile strength?

Author Response

(The authors gave the same response as above.)

Round 2

Reviewer 1 Report

This manuscript has been accepted in this present form.

Author Response

We would like to thank the Reviewer for detailed comments and suggestions that helped to improve the manuscript. 

Reviewer 3 Report

The reviewer has no further comment.

Author Response

(The authors gave the same response as above.)
